# Microbiome and Prostate Cancer: Emerging Diagnostic and Therapeutic Opportunities

**DOI:** 10.3390/ph17010112

**Published:** 2024-01-13

**Authors:** Sung Jin Kim, Myungchan Park, Ahnryul Choi, Sangjun Yoo

**Affiliations:** 1Department of Urology, Gangneung Asan Hospital, University of Ulsan College of Medicine, Gangneung 25440, Republic of Korea; bop1004@ulsan.ac.kr; 2Department of Urology, Haeundae Paik Hospital, Inje University College of Medicine, Busan 47392, Republic of Korea; mcpark@paik.ac.kr; 3Department of Biomedical Engineering, College of Medical Convergence, Catholic Kwandong University, Gangneung 25601, Republic of Korea; 4Department of Urology, SNU-SMG Boramae Medical Center, Seoul 07061, Republic of Korea

**Keywords:** prostate cancer, urine microbiome, gut microbiome

## Abstract

This review systematically addresses the correlation between the microbiome and prostate cancer and explores its diagnostic and therapeutic implications. Recent research has indicated an association between the urinary and gut microbiome composition and prostate cancer incidence and progression. Specifically, the urinary microbiome is a potential non-invasive biomarker for early detection and risk evaluation, with altered microbial profiles in prostate cancer patients. This represents an advancement in non-invasive diagnostic approaches to prostate cancer. The role of the gut microbiome in the efficacy of various cancer therapies has recently gained attention. Gut microbiota variations can affect the metabolism and effectiveness of standard treatment modalities, including chemotherapy, immunotherapy, and hormone therapy. This review explores the potential of gut microbiome modification through dietary interventions, prebiotics, probiotics, and fecal microbiota transplantation for improving the treatment response and mitigating adverse effects. Moreover, this review discusses the potential of microbiome profiling for patient stratification and personalized treatment strategies. While the current research identifies the pivotal role of the microbiome in prostate cancer, it also highlights the necessity for further investigations to fully understand these complex interactions and their practical applications in improving patient outcomes in prostate cancer management.

## 1. Introduction

Over 1.4 million new patients were diagnosed with prostate cancer, and over 370,000 patients died of prostate cancer worldwide in 2020 [1]. Prostate cancer is commonly recognized as a hereditary condition, with genetic variations that significantly influence its development risk and susceptibility [2]. Lifetime prostate cancer risk can be accurately estimated using a polygenic risk score incorporating more than 269 germline risk variants [3]. Despite advancements in the understanding of prostate cancer through genome-wide studies, their impact on altering diagnostic and treatment approaches for the disease remains relatively limited. In addition to genetic factors, lifestyle elements, such as diet and physical activity, are considered modifiable risk factors for prostate cancer and its aggressive forms [4]. However, the precise relationship between specific lifestyle factors and prostate cancer risk still requires further clarification [5,6,7].

The Human Microbiome Project is a conceptual extension of the Human Genome Project, which aims to elucidate the entire human genetic landscape and includes human cells as well as the microbial cells residing inside and on humans [8]. In contrast to the fixed nature of the genome, the human microbiome is dynamic and susceptible to alterations that can undergo changes in response to lifestyle changes. Such shifts in the microbiome can cause dysbiosis, a state of microbial imbalance affecting health status and predisposing individuals to various diseases, including different types of cancer. Genetic and lifestyle factors affect prostate cancer development and aggressiveness, and the microbiome reflects both these factors. The microbiome is also crucial to uncovering previously unexplored aspects of prostate cancer prevention, diagnosis, risk assessment, and treatment strategies.

Prostate cancer remains a significant global health challenge. Its etiology and progression are influenced by various factors, including genetics, lifestyle, and microbiome composition. Despite the known impact of the microbiome on health and disease, its role in prostate cancer remains underexplored. Moreover, although recent research has begun to focus on the microbiome’s role in prostate cancer, the practical applications of these findings in diagnosis and treatment have not been sufficiently elucidated. This highlights the need to focus on how the microbiome can be integrated into clinical practice. In this review, we explore the potential uses of the microbiome and examine the limitations of urinary and gut microbiomes in various aspects of prostate cancer management, including prevention, diagnosis, risk stratification, and treatment. Our investigation aligns with recent calls for more in-depth research in this domain.

## 2. Evidence Acquisition

Four independent authors (S.J.K., M.P., A.C., and S.Y.) systematically reviewed the studies published in English using the PubMed database or Google Scholar through October 2023. The following search terms were used: (microbiota), (microbiome), (cancer), (neoplasms), (prostate cancer), or (prostate neoplasm). Additionally, published articles cited in the studies identified using the above methods were selected.

## 3. Direct Microbiome in Prostate Cancer

### 3.1. Urinary Microbiome and Prostate Cancer

Urine has traditionally been regarded as sterile, and this old dogma has created a conceptual barrier to active research on the urinary microbiome [9]. Moreover, urine was not included in the Human Microbiome Project because of this dogma [8]. Consequently, the urine microbiome in healthy adults is not well documented, and defining a normal urine microbiome is difficult [10]. Therefore, research on the urinary microbiome has received little attention. However, some studies have reported promising results using the urinary microbiome, particularly in lower urinary tract symptoms (LUTs) and inflammatory disease [11], and some recent studies have focused on the relationship between prostatic diseases, including prostate cancer, and the urinary microbiome.

Although the relationship between prostate cancer and the urinary microbiome is not well documented, the microbiome may be a key factor that induces chronic prostate inflammation. Infection and inflammation in the prostate, which are related to the prostatic microbiome, lead to microenvironmental changes in the prostate and promote prostatic carcinogenesis [12,13,14]. Previous studies have shown that the prostate microbiome is similar to that of the urethra, supporting bacterial translocation to the prostate through the urethra [15,16]. In other words, asymptomatic chronic prostate inflammation can be induced by the urinary microbiome, which may be a notable cause of prostate cancer development. Intraprostatic urine reflux must be considered to further understand the relationship between the urinary microbiome and prostatic inflammation [17,18,19]. Generally, men void 5–7 times daily, and every single voiding event spreads the urinary microbiome into the prostate [15,16]. Although the urine biomass is extremely low, these frequent daily events would cause asymptomatic prostatic microenvironmental changes and chronic inflammation. Furthermore, pre-existing urinary dysbiosis can cause prostatic dysbiosis [10,20,21]. Although direct prostatic microbiome acquisition would be the best option for assessing the relationship between the microbiome and prostate cancer, such studies cannot be conducted easily because of their invasiveness and contamination risk. Considering these limitations, urine is one of the most easily accessible and reasonable specimens for microbiome research in prostate cancer.

### 3.2. Recent Studies of the Urinary Microbiome in Prostate Cancer

Table 1 lists recent studies on the urinary microbiome of patients with prostate cancer. Initial investigations into the role of the urinary microbiome in prostate cancer have yielded varied results. Yu et al. (2015) reported a significant increase in the abundance of *Bacteroidetes*, *Alphaproteobacteria*, *Firmicutes*, *Lachnospiraceae*, *Propionicimonas*, *Sphingomonas*, and *Ochrobactrum* in the prostatic secretions of patients with prostate cancer compared to those of patients with benign prostatic hyperplasia (BPH) [22]. Shrestha et al. identified a bacterial cluster, including *Streptococcus anginosus*, *Anaerococcus lactolyticus*, *Anaerococcus obesiensis*, *Actinobaculum schaalii*, *Varibaculum cambriense*, and *Propionimicrobium lymphophilum*, associated with an increased prostate cancer risk (70.8% vs. 46.7%, *p* = 0.041) [23]. This finding suggests a potential link between these organisms, urogenital infections, and prostate cancer risk.

In contrast to previous studies, Alanee et al. (2019) collected the first voided urine sample after prostate massage and observed an increased abundance of *Veillonella*, *Streptococcus*, and *Bacteroides*, along with a decreased abundance of *Faecalibacterium*, *Lactobacilli*, and *Actinobacter* in patients with prostate cancer compared to samples from healthy patients [24]. Tsai et al. (2022) revealed distinct differences in the urinary microbiome of individuals with LUTs, BPH, and prostate cancer, with the males with LUTs without BPH as a control group [25]. In prostate cancer patients, *Faecalibacterium*, *Staphylococcus*, *Ruminococcaceae_UCG_002*, *Neisseria*, and *Agathobacter* were more prevalent than in healthy participants. Additionally, compared to the bacterial frequencies in urine samples from patients with BPH, *Escherichia*, *Shigella*, *Sphingomonas*, *Subdoligranulum*, *Blautia*, and *Pseudomonas* were more frequent in samples from patients with prostate cancer. Furthermore, Hurst et al. (2022) reported a notable correlation between the urinary microbiome and increased prostate cancer risk, along with traditional indicators such as prostate-specific antigen (PSA) levels, Gleason scores, and clinical stages [26]. This study identified a set of anaerobic bacteria, including *Fenollaria*, *Peptoniphilus*, *Porphyromonas*, *Anaerococcus*, and *Fusobacterium*, potentially associated with poor prognosis in prostate cancer.

Human papillomavirus (HPV), a sexually transmitted virus, causes various types of cancers, including cervical, head, and neck cancers [27]; however, its direct role in prostatic carcinogenesis is still not well understood. Some studies have found a difference in the prevalence of HPV infection between prostate cancer patients and healthy participants [28,29,30,31,32], whereas others have found no difference [33,34,35,36]. The effect of HPV on prostate cancer may be linked to various aspects of carcinogenesis, including immune system regulation, hormone metabolism, and gene damage [37,38,39,40]. Evidence suggests that high-risk HPV (types 16 and 18) immortalizes normal prostate cells and transforms them into prostate cancer cells [38]. In another study involving the same patients, high-risk HPVs were detected in prostate tissue prior to prostate cancer diagnosis [37]. HPV may indirectly inhibit the protective function of APOBEC3B against other viral infections, and HPV E7 may be directly involved in prostate carcinogenesis [39,40]. However, because of the low prevalence of HPV infection, conflicting research results have been published, even within the same region [32,33], and further studies are required to prove the causal relationship between HPV infection and prostate cancer development. However, considering the efficacy of the HPV vaccine, population-based longitudinal studies should be conducted to determine whether HPV vaccination can effectively prevent prostate cancer.

Several studies have reported that males with human immunodeficiency virus (HIV) have a decreased incidence of prostate cancer compared to males without HIV [41]. These findings may be due to the effects of antiretroviral therapy, which reportedly reduces cancer risk [42], as well as because many males with HIV die from other causes before developing prostate cancer. Infections caused by various fungi, including *Candida albicans*, *Blastomycosis*, and *Cryptococcus*, are rare causes of prostatitis, particularly in HIV-positive males. The metabolic products of these fungi, known as mycotoxins, are carcinogenic [43,44]; however, a direct relationship between these infections and prostate cancer has not yet been established [45].

### 3.3. Potential Pharmaceutical Applications of Urinary Microbes and Future Direction

A potential hypothesis for how the urine microbiome is associated with prostate carcinogenesis is that when these pro-inflammatory species infect the prostate, they may cause chronic inflammation that promotes proliferative inflammatory atrophy, which is a risk factor for prostate cancer [12]. The difficulty in determining the role of the urinary microbiome in prostate cancer development is biased due to not only the problem of hard-to-cultivate species but also our limited knowledge about the interactions among bacteria, as well as between the host and bacteria. Based on the strongly connected general interactions between bacteria and their hosts, several urinary microbes may contribute to prostate carcinogenesis [12]. However, because most studies on the urine microbiome have reported results focusing on the differences between patients with and without prostate cancer, a longitudinal study design is needed to determine the causal relationship between the urinary microbiome and prostate cancer development. Additionally, the existence of the urinary microbiome identified in previous studies on prostate cancer tissues should be confirmed using a longitudinal study design. Moreover, most current studies have focused on the relationship between specific urinary microbes and prostate cancer development. Unfortunately, the results are not homogeneous, and various microbe species are associated with prostate cancer diagnosis. In other words, we hypothesized that the existence of specific microbiomes, as well as overall changes in the microbial community in the urinary tract, known as dysbiosis itself, could be the microbial status that we need to focus on in prostate cancer research.

Only few published studies have focused on the urinary microbiome in prostate cancer. Moreover, although prostate cancer patients showed distinguishable urinary microbiome characteristics compared to those with BPH or healthy participants, these studies reported different results. However, considering the high proportion of negative results after prostate biopsy based on PSA levels [46] or prebiopsy magnetic resonance imaging [47], novel biomarkers for prostate cancer are urgently required. Additionally, the prediction of lethal prostate cancer using genetic risk is helpful in improving public health, although its accuracy requires improvement [48]. In this clinical situation, the urinary microbiome may serve as a potential biomarker to improve prostate cancer prediction and risk calculations. Additionally, the urinary microbiome could also be helpful in accurately assessing prostate cancer aggressiveness [26].

However, the heterogeneity of the results regarding the influence of the urinary microbiome on prostate cancer may be suspicious. In other words, the urinary microbiome found in prostate cancer patients could be a result of contamination, and the association of the urinary microbiome with other factors, such as the prostate volume, urinary tract symptoms, or lifestyle-related variables, cannot be ruled out completely.

Although these questions remain difficult to answer, and more research is needed, a recent study could provide a possible explanation for the causality between the urinary microbiome and prostate cancer development. A previous study reported interesting results that theoretically supported the development and progression of the urinary microbiome and prostate cancer development and progression [49]. Urinary microbes can metabolize glucocorticoids for energy and subsequently generate androgens, leading to increased prostate cancer cell proliferation and cancer progression. These results support the possible causality between the urinary microbiome and prostate cancer development and aggressiveness. Considering these results, future studies on the metabolic characteristics of each urinary microbe should be conducted. In particular, the finding that the urinary microbiome differs according to prostate cancer diagnosis should be helpful for lifetime prostate cancer risk calculations.

### 3.4. Limitations to Urinary Microbiome Use in Prostate Cancer Research

Before mentioning the recent studies, the urine collection methods should be briefly summarized to further understand the limitations of using urinary microbiome in prostate cancer research. Suprapubic aspiration may be the best method for urine specimen collection to minimize the probability of contamination; however, this method still has the probability of contamination from the skin [50]. Moreover, only few studies have obtained urine specimens using suprapubic aspiration methods, owing to their invasiveness [51,52]. Most studies have focused on the urinary microbiome using specimens obtained from midstream voided urine or catheterization. Therefore, the results from studies focusing on the urinary microbiome must be carefully interpreted, and the contamination probability must be considered. In fact, when looking at the summarized results, the urinary microbiome shows considerable similarities with the gut or vaginal microbiomes [53]. Even if these results are not due to contamination, they may require caution during interpretation.

Moreover, many studies focusing on prostate cancer obtained urine specimens after digital rectal examination or prostatic massage [54]. A recent study has reported that these procedures can alter the urinary microbiome [23,55]. Additionally, owing to the low biomass of urine, the microbiome presents in instruments used during specimen collection, such as catheters, lubricating jelly, or materials used in the experiments, such as DNA preservatives, could be problematic; however, these poorly impact fecal microbiome studies. Therefore, to reduce the probability of misinterpretation in urine microbiome research, attention should be paid to the procedures before urine collection and the urine collection methods used in these studies.

## 4. Indirect Microbiome in Prostate Cancer

### 4.1. Gut Microbiome and Prostate Cancer

Although the prostate is not directly associated with the gut, prostate cancer is closely associated with the gut microbiome. Traditionally, prostate cancer is affected by lifestyle factors, which can subsequently alter the gut microbiome (Figure 1). Additionally, the gut microbiome and its metabolites are involved in prostate cancer development [56]. Moreover, recent studies have revealed that the gut microbiome is associated with several diseases, including various cancers that are not directly associated with the gut [57,58]. However, though the association between the gut microbiome and digestive tract cancer has been extensively studied [59], few studies have focused on the relationship between the gut microbiome and prostate cancer. Additionally, owing to their indirect association and the relatively low contamination risk owing to high stool biomass, precisely elucidating the pathophysiological pathway linking the gut microbiome and prostatic disease compared to the directly connected urinary microbiome is challenging. Despite these limitations, studies focusing on the association between prostate cancer and gut microbiome may prove particularly helpful when considering novel gastrointestinal tract-delivered drugs for prostate cancer.

The identification of reliable, non-invasive biomarkers for early detection and risk assessment in prostate cancer is valuable. The urinary and gut microbiomes hold significant potential in this regard, and recent research supports the diagnostic capabilities of these microbiomes. Increasing understanding of the relationships among different microbiomes and prostate cancer has revealed that cancerous prostate tissue, unlike healthy prostate tissue, contains bacterial DNA. Moreover, the urinary microbiome, when linked to chronic inflammation, has been associated with an increased risk of prostate cancer and its pathogenesis, further suggesting its potential as a diagnostic tool [60]. Several studies have explored differences in the microbiota of patients with prostate cancer. In addition, dysbiosis may affect the inflammatory status and hormone levels of the body and result in reduced microbial metabolite levels, leading to prostate cancer progression. This provides a basis for establishing the gut microbiome as a biomarker for early detection and assessment of prostate cancer risk [61]. Furthermore, the role of the gut microbiome in prostate cancer endocrine resistance, which occurs through the generation of androgenic steroids, also supports its potential diagnostic capability. Its composition could potentially reveal the endocrine resistance status of a patient with prostate cancer [62].

### 4.2. Recent Studies on Gut Microbiome Involvement in Prostate Cancer

The gut microbiome is associated with prostate cancer risk through obesity and inflammation, often attributed to metabolite-induced prostate carcinogenesis. However, the underlying mechanism remains to be fully elucidated. Although some studies have aimed to reveal the association between the gut microbiome and prostate carcinogenesis, only a few investigated differences in the gut microbiome between patients with prostate cancer and healthy participants (Table 2) [24,63,64,65]. Liss et al. (2018) reported that *Bacteroides* and *Streptococcal* species were more prevalent in patients with prostate cancer than in healthy individuals; they used rectal swabs for sampling before prostate biopsy [63]. Golombos et al. (2018), using stool samples obtained before prostate biopsy, reported that *Bacteroides* spp. were more prevalent in patients with prostate cancer, whereas *Faecalibacterium prausnitzii* and *Eubacterium* spp. were more prevalent in patients without prostate cancer [64]. Alanee et al. (2019) reported that *Bacteroides* spp. are prevalent in patients with prostate cancer [60]. In 2021, Matsushita et al. reported a significant increase in *Rikenellaceae*, *Alistipes*, and *Lachnospira* spp. abundance in patients with aggressive prostate cancer [65]. In a preclinical model, this study showed that short-chain fatty acids (SCFAs) and gut bacterial metabolites promoted prostate cancer growth through insulin growth factor-1 signaling. SCFAs are metabolites produced by the gut bacteria through dietary fiber fermentation; they regulate the immune system and exert anti-inflammatory effects, potentially playing a role in prostate cancer prevention.

In the human host, Gram-negative bacteria, as well as commensal anaerobic bacteria, are normal residents of the gut microbiome. Lipopolysaccharides (LPSs), also known as endotoxins, are components of the outer membranes of Gram-negative bacteria. LPSs contribute to prostate carcinogenesis through toll-like receptor-4 activation [66] and increase blood vessel permeability, which supports cancer cell extravasation, as prostate cancer metastasis through NF-κB and inflammatory cytokine upregulation [67]. Weight gain, a high-fat diet, and fatty acid exposure, which are risk factors for prostate cancer and weaken the gut barrier, promote LPS translocation, which might be a potential link between prostate carcinogenesis and the gut microbiome [68,69]. Owing to the diverse roles of inflammation in cancer progression, anti-inflammatory drugs have become a treatment for controlling inflammation-induced metastasis, particularly in castration-resistant prostate cancer (CRPC) patients [70].

### 4.3. Recent Studies on Gut Microbiome Involvement in Prostate Cancer Treatment

The microbiome is composed of mutually beneficial bacteria that live in close contact with their host and that are mutually beneficial [71]. However, carcinogenesis and cancer treatment can cause dysbiosis through multiple mechanisms. Additionally, the microbiome modulates the response to cancer chemotherapy and immunotherapy [72] and is intimately implicated in the biotransformation of anticancer medicines, which causes unintended effects in cancer treatment [73]. Moreover, the gut microbiome can alter androgen levels, which are strongly associated with prostate cancer development and treatment [74].

**Table 2 pharmaceuticals-17-00112-t002:** Recent studies of the gut microbiome in prostate cancer.

Authors	Published Year, Reference	Study Design	Groups (n)	Materials	Method	Main Pathogens Found
Liss et al.	2018, [63]	Prostate biopsy	PCa (64) vs. non-PCa (41)	Rectal swabs	16S rRNA amplicon sequencing	High in PCa: *Bacteroides Streptococcus*
Golombos et al.	2018, [64]	Prostate biopsy	PCa (20) vs. non-PCa (8)	Stool samples before	16S rDNA amplicon sequencing	High in PCa: *Bacteroides massiliensis*, Controls: *Faecalibacterium prausnitzii*, *Eubacterium rectalie*
Sfanos et al.	2018, [75]	PCa vs. Control	NoMeds (16) vs. ADT (5) vs. AA	Rectal swabs	16S rDNA amplicon sequencing	High in ADT: *Akkermansia muciniphila*, *Ruminococcaceae* spp., *Lachnospiraceae* spp., AA: *Brevibacteriaceae* (low), *Erysipelorichaceae* (low), *Streptococcaceae* (low)
Alanee et al.	2019, [24]	Prostate biopsy	PCa (30) vs. non-PCa (16)	Rectal swabs	16S rRNA amplicon sequencing	High in PCa: *Bacteroides*
Daisley et al.	2020, [76]	PCa vs. Control	ADT (21) vs. ADT + AA (14) vs. control (33)	Stool samples	16S rRNA amplicon sequencing	ADT: *Corynebacterium* (low) High in ADT + AA: *Akkermansia*
Liu et al.	2020, [77]	PCa	HSPC (21) vs. CRPC (21)	Stool samples	16S rRNA amplicon sequencing	High in CRPC: *Phascolarctobacterium* and *Ruminococcus*
Matsushita et al.	2021, [65]	Prostate biopsy	PCa (30) vs. non-PCa (16)	Rectal swabs	16S rRNA amplicon sequencing	High in PCa: *Rikenellaceae*, *Alistipes*, and *Lachnospira*
Pernigoni et al.	2021, [78]	PCa	HSPC (19) vs. CRPC (55)	Rectal swabs	16S rDNA amplicon sequencing	High in HSPC: *Prevotella stercorea* High in CRPC: *Ruminococcus* spp., *Bacteroides* spp.

PCa, prostate cancer; ADT, androgen deprivation therapy; AA, abiraterone acetate; HSPC, hormone-sensitive prostate cancer; CRPC, castration-resistant prostate cancer; spp., species.

Table 2 lists recent studies on the microbiome of patients with prostate cancer and those receiving androgen-axis-associated therapy [75,76,77,78]. Recent evidence demonstrates that the gut microbiome participates in the host androgen metabolism and may be associated with prostate cancer development, progression, and treatment [79]. The gut microbiome can modulate the circulating androgen amount by influencing host cells, directly biotransforming and synthesizing them. Additionally, a bacterium related to the testosterone biosynthetic pathway and increasing testosterone levels during androgen deprivation therapy (ADT) has been identified [75]. Moreover, CRPC development may also be related to this testosterone biosynthesis [77,78]. Sfanos et al. (2018) reported an increased abundance of the gut microbiome, including *Akkermansia* spp., *Ruminococcus* spp., and *Lachnospiraceae* spp., which are related to the expression of the bacterial gene pathways involved in androgen biosynthesis in prostate cancer patients [75]. Liu et al. (2020) have reported an increase in *Ruminococcus* spp. and *Phascolarctobacterium* abundance in CRPC patients compared to that in hormone-sensitive prostate cancer (HSPC) patients [77]. Moreover, Daisley et al. (2020) found that treatment with abiraterone acetate was associated with an increased abundance of *Akkermansia muciniphila*, which might contribute to the therapeutic effects of abiraterone acetate in CRPC patients [76]. Pernigoni et al. (2021) demonstrated an increased *Ruminococcus* spp. and *Bacteroides* spp. abundance in mice and humans with CRPC. These species are related to androgen production, providing evidence that the gut microbiome interferes with complete androgen suppression during ADT. Additionally, the gut microbiome may play a substantial role in prostate cancer progression from HSPC to CRPC [78].

### 4.4. Potential Therapeutic Application of the Gut Microbiome

Several epidemiological studies have indicated a correlation between specific urinary and gut microbiota compositions and an increased risk of prostate cancer. Additionally, as previously mentioned, various studies have presented mechanistic evidence linking the microbiome to prostate cancer, suggesting its potential as a biomarker. The role of the gut microbiota in endocrine resistance in prostate cancer has been substantiated, at least partially, through the generation of androgenic steroids that facilitate carcinogenic signaling via androgen receptors [78]. Furthermore, the production of bacterial genotoxins is associated with the occurrence of TMPRSS2-ERG gene fusion, a common early carcinogenic event during the development of prostate cancer. Therefore, the host microbiota provides mechanistic evidence relating to the onset and progression of prostate cancer, suggesting that the profiling and modulation of this microbiome could offer potential avenues for detection and treatment [62]. Intriguingly, the gut microbiota profile itself has emerged as a novel and useful marker for the detection of high-risk prostate cancer, further implicating the microbiome in the carcinogenesis of this cancer type [65]. Alterations in the composition of the microbiota are essential factors in the occurrence, development, and prognosis of prostate cancer; once again suggesting that the microbiome may be an innovative biomarker [80]. Besides providing insights into the mechanisms of the microbiota’s influence on prostate cancer development and on its potential as a biomarker, the gut microbiota is involved in the progression and chemoresistance of prostate cancer cells [81]. Furthermore, studies have shown that a high-fat diet leads to gut dysbiosis and promotes prostate cancer growth through bacterial metabolites [82]. Comprehensive studies profiling different microbiomes —including those of prostate tissue, the gut, urine, seminal fluid, and prostatic fluid— in relation to prostate cancer provide an extensive understanding of the microbiome composition diversity and its association with the disease. Such studies are invaluable in supporting the microbiome as a potential biomarker and delineating the multifaceted roles of microorganisms in prostate cancer pathogenesis [83]. Additionally, the presence of proinflammatory or pathogenic bacteria, particularly those involved in urinary tract pathologies, has been linked to the progression of prostate cancer, highlighting the role of the microbiome in prostate cancer pathogenesis and further endorsing its potential as a biomarker and a target for therapeutic strategies [53].

Innovative therapeutic strategies are crucial in advancing prostate cancer treatment. Recent studies have explored the potential of various therapeutic strategies, including specific dietary interventions, the ingestion of probiotics and/or prebiotics, and fecal microbiota transplantation (FMT) for microbiome modification, aimed at improving treatment responses and developing personalized therapies for patients with prostate cancer. This study explores the use of probiotics as a potential adjuvant therapy for prostate cancer. It suggests that probiotics might enhance the response to treatment and reduce the risk of postoperative infections, indicating the potential of microbiome-related strategies in prostate cancer therapy [84]. Further, a study showed that the gut microbiome plays a role on the therapeutic efficacy of prostate cancer immunotherapies —including cyclophosphamide and anti-PD-L1 therapies, and CTLA-4 blockade— and suggested that microbiome modulation may be a potential therapeutic strategy for prostate cancer [85]. The emerging evidence on the participation of the gut microbiome in health and disease suggests that alterations in its composition (dysbiosis) may play a crucial role in the occurrence, development, and prognosis of prostate cancer, highlighting the potential of microbiome modification as a novel therapeutic strategy [80].

Several potential treatment methods for gut microbiome dysbiosis, including the use of probiotics, prebiotics, antibiotics, bioactive metabolites, bacteriophages, and fecal microbiota transplantation, have been reported in prostate cancer patients. These methods aim to modify the gut microbiome composition and function to improve health outcomes in prostate cancer patients. One approach is the use of probiotics, live microorganisms that can confer health benefits to the host, which improve the gut microbial balance and decrease inflammation, making them a potential treatment for prostate cancer. Some studies have shown that certain bacterial strains, such as *Lactobacillus*, can reduce inflammation and improve immune function, which may be beneficial for individuals with prostate cancer [86]. Another approach involves the use of prebiotics, which are non-digestible food ingredients that stimulate the growth of beneficial microorganisms in the gut. Prebiotics have anti-inflammatory and anticancer properties and may be useful in prostate cancer treatment. FMT, which involves transplanting healthy gut bacteria from a donor into a patient to increase beneficial gut microbiota, is another approach used to treat gut microbiome dysbiosis in prostate cancer patients.

Sex determines the composition of the gut microbiome, which also participates in the hormone metabolism associated with prostate cancer development and treatment [79]. In a recently identified mechanism, an ADT-induced host androgen imbalance can be interfered via gut microbiome-induced androgen. In other words, gut-microbiome-induced androgens could lead to treatment failure after ADT and, eventually, CRPC development [78]. Targeting the crosstalk between the host and bacteria through the direct involvement of the gut microbiome or its metabolic pathways in androgen biosynthesis may represent an alternative therapeutic strategy for patients with advanced prostate cancer. In addition to the effect of antibiotics on the androgen-related gut microbiome in this study, *Prevotella stercorea* inhibited CRPC tumor growth in castrated mice by competing for the expansion of androgen-related bacteria, suggesting the possibility of a treatment utilizing FMT [78].

The modulation of the microbiota during immune checkpoint blockade therapy can improve oncological outcomes in cancer patients. Several studies have identified beneficial bacteria through cancer treatment responses and improved responses to immunotherapy in mice transplanted with these bacteria. In patients with melanoma treated with immune checkpoint blockade therapy, the *Ruminococcaceae* spp. abundance increased in the well-responsive group [87]. In patients with epithelial tumors, *A. miciniphila* abundance increased in those with a favorable response to immune checkpoint inhibitors [88]. Prostate cancer patients who received ADT showed an abundance of the gut microbiome (*A. muciniphila* and *Ruminococcaceae* spp.), which is associated with improvements in immune checkpoint inhibitor therapy compared to patients who did not receive ADT [75]. The concordance of microorganisms between patients responding favorably to immune checkpoint blockade therapies and ADT may indicate the presence of bacteria that support cancer treatment and serve as the foundation for therapeutic approaches, such as FMT.

### 4.5. Emerging Insights and Future Directions of the Gut Microbiome in Prostate Cancer Research

In recent studies, the perspectives on the microbiome are not only innovative but also provide comprehensive insights on the roles played by the different microbiomes within the body. These perspectives emphasize the significant impact of host-microbiome interactions on multiple physiological processes and a range of multifactorial disease conditions [59,89,90,91,92]. The role of commensal bacteria and other microorganisms that colonize the epithelial surfaces of the body is a subject of intense study. The production of small molecules and metabolites by these microorganisms underscores the intricate relationship between the microbiome and cancer, demanding further exploration [59]. The literature frequently discusses host mechanisms by which the microbiome may modify carcinogenesis, thus offering a comprehensive understanding of the microbiome’s role in cancer and its potential implications for diagnosis and therapy [89]. Additionally, the use of the salivary microbiome, fecal microbiome, and circulating microbial DNA in blood plasma as experimental diagnostic biomarkers for numerous types of cancer supports the potential diagnostic capabilities of the microbiome [90]. Furthermore, the human microbiome, recognized as a complex community, engages in symbiotic interactions with the host across multiple body sites [91]. The impact of microorganisms on cancer immunotherapy proposes a new direction for improving the effects of tumor immunotherapy through the application and/or manipulation of microorganisms. It provides insights into carcinogenic microorganism infections, microbial disorders, and carcinogenesis, offering a comprehensive view of the role of the microbiome [92].

Oxidative stress is a state of imbalance between the production of free radicals and the ability of the body’s antioxidant defense system to neutralize them. Environmental-pollutant-induced free radicals can damage cellular components, including DNA, leading to oxidative stress and the development of cancers, including prostate cancer [12,93,94]. The use of nano-antioxidants that target the delivery of nano-vaccines to dendritic cells (DCs) via DC-binding peptides induces potent antiviral immunity in vivo [95]. It may protect against oxidative damage and reduce the carcinogenesis risk, suggesting that it could be a potential strategy for the prevention and treatment of cancers, including prostate cancer [93]. The use of antioxidants for prostate cancer remains a new field of research, and more work is needed to evaluate the mechanisms of action and optimal use of these treatments. However, the results of these studies suggest that nano-antioxidants may be used as therapeutics for the existing commercial antibodies or a source of new drugs for prostate cancer treatment.

Several microbiome-produced bioactive metabolites have effects on prostate cancer. Bile acids and indoles are among the most important bioactive metabolites. Some studies have suggested that alterations in the microbiome can change the bile acid metabolism, which may contribute to prostate cancer development [96,97]. Indoles are gut bacterial metabolites with anti-inflammatory and anticancer effects. Some studies have suggested that indoles play a role in prostate cancer treatment.

The gut microbiome differs according to complex factors such as lifestyle, dietary habits, genetic background according to race, and individual susceptibility. Thus, determining the association between the gut microbiome and prostate cancer remains difficult after adjusting for these variables. Moreover, how the gut microbiome detects abnormal androgen levels in the host, and how it expands androgen-related species, remains unclear. Although differences in the composition of the gut microbiome have been identified in prostate cancer, the pathogenesis remains unknown. Recent evidence on the role of the gut microbiome in prostate cancer could be expanded to treatment methods such as FMT or a commercial probiotic; however, this aspect requires further research.

## 5. Limitations of the Studies

Several limitations exist in these urinary and gut microbiome studies. In urinary microbiome research, the studies have primarily focused on the differences between patients with and without prostate cancer. However, the results concerning the types of microorganisms constituting the microbiome are often inconsistent. This inconsistency may stem from varied study methodologies, including urine collection and processing techniques, which can lead to contamination and affect the outcomes. Moreover, the causal relationship between urinary microbiome alterations and prostate cancer development remains unclear. The temporal gap between the typical age of occurrence of urinary tract infections and the age of prostate cancer onset further complicates our understanding of any potential causal links, suggesting a need for prospective longitudinal studies to elucidate this relationship. Regarding the gut microbiome, the research has primarily focused on the differences between patients with and without prostate cancer, as well as between patients at low- and high-risk of developing prostate cancer. This approach is one of the most relevant methods for understanding the complex interplay between the gut microbiome and prostate cancer, but many aspects of the causal relationships remain to be elucidated. Recent studies have indicated that the gut microbiome might regulate testosterone levels, which are crucial in prostate cancer treatment. These findings suggest that the gut microbiome interferes with testosterone deprivation and could be linked to treatment failures. In addition, the evidence from these studies provides a basis for understanding the causal links. Moreover, FMT has shown potential as a treatment method in these experimental settings, but its effectiveness typically requires continuous application. Therefore, the clinical efficacy of other possible treatments needs further exploration. Besides, the use of prebiotics and probiotics necessitates additional research involving human subjects to validate their therapeutic potential and establish a more concrete basis for their use in prostate cancer management.

## 6. Conclusions and Future Directions

The role of urinary and gut microbiomes in prostate cancer improves our understanding of cancer pathogenesis and management. The potential of the urinary microbiome as a non-invasive biomarker allows early detection and risk assessment in prostate cancer, while the gut microbiome’s impact on the efficacy of treatments such as chemotherapy, immunotherapy, and hormone therapy holds promise for improved patient outcomes. However, the work in this field is still in its early stages, and further research is needed to validate the clinical utility of these microbiome profiles.

Future studies should focus on standardizing microbiome analysis methodologies and consider factors, such as diet, lifestyle, and genetics, that can influence microbiome composition. Additionally, how the microbiome affects prostate cancer progression and treatment responses, including dietary intervention and microbiota modulation strategies, should be explored. With the advances in research, individual microbiome profile-based personalized medicine approaches have become a tangible goal, potentially revolutionizing prostate cancer diagnosis and treatment, and thus enhancing patient care.

## Figures and Tables

**Figure 1 pharmaceuticals-17-00112-f001:**
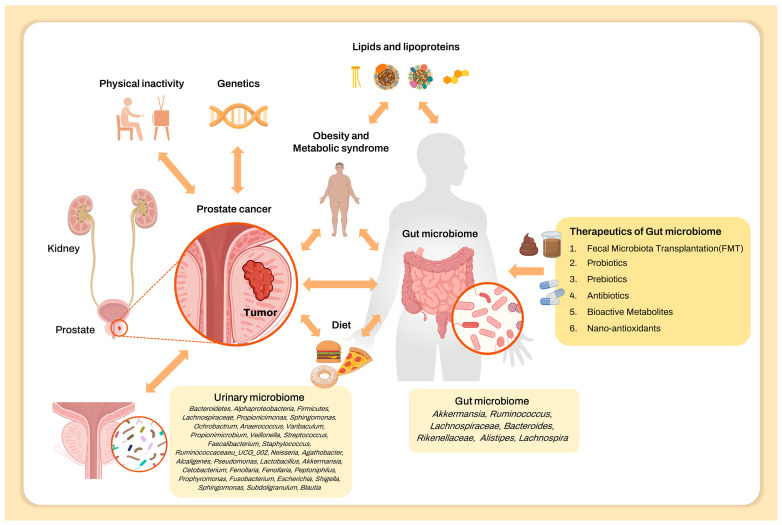
Risk factors including urinary and gut microbiome related to the development of prostate cancer and therapeutics of the gut microbiome.

**Table 1 pharmaceuticals-17-00112-t001:** Recent studies of urinary microbiome in prostate cancer.

Authors	Published Year, Reference	Groups (n)	Urine Sample	Study Design	Main Pathogens Found
Haining Yu et al.	2015, [22]	BPH (21) PCa (13)	Expressed prostatic secretions	PCa vs. BPH	*Bacteroidetes*, *Alphaproteobacteria*, *Firmicutes*, *Lachnospiraceae*, *Propionicimonas*, *Sphingomonas*, *Ochrobactrum*
Eva Shrestha et al.	2018, [23]	Non-PCa (65) PCa (repeat Bx) (5)	Not mentioned	PCa vs. non-PC	*Anaerococcus lactolyticus*, *Varibaculum cambriense*, *Propionimicrobium lymphophyilum*
Shaheen Alanee et al.	2019, [24]	non-PCa (16) PCa (14)	First voided urine after prostate massage	PCa vs. non-PCa	*Veillonella*, *Streptococcus*, and *Bacteroides* (increased) *Faecalibacterium*, *Lactobaccili*, and *Actinetobacter* (decreased)
Kai-Yen Tsai et al.	2022, [25]	BPH (77) nmPCa (59) Control (36)	Mid-term voided urine	PCa vs Control PCa vs BPH	*Faecalibacterium*, *Staphylococcus*, *Ruminococcaceaeu*_UCG_002, *Neisseria*, *Agathobacter*, *Alcaligenes*, *Pseudomonas*, *Lactobacillus*, *Akkermansia*, *Cetobacterium*
Rachel Hurst et al.	2022, [26]	PCa w/u (300) Hematuria (18)	First voided urine after DRE	PCa vs. non-PCa (Risk/Prognosis)	*Fenollaria*, *Fenollaria*, *Peptoniphilus*, *Prophyromonas*, *Anaerococcus*, and *Fusobacterium*

PCa, prostate cancer; DRE, digital rectal examination; BPH, benign prostate hyperplasia; nmPCa, non-metastatic prostate cancer.

## Data Availability

Data sharing is not applicable.

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
