# Peer review of "Microbiome and Prostate Cancer: Emerging Diagnostic and Therapeutic Opportunities"

_pharmaceuticals, 2024, doi:10.3390/ph17010112_

Round 1

Reviewer 1 Report

Comments and Suggestions for Authors

The manuscript entitled “Microbiome and Prostate Cancer: Emerging Diagnostic and Therapeutic Opportunities” is a narrative review which focuses on the advantages and limitations of prostate microbiome profiling as a diagnostic tool, as well as on therapeutic potential of novel strategies targeting microbiota. The topic of the manuscript is within the scope of the journal and would be interesting to the readership focusing on cancer biology, cancer biomarkers and innovative cancer therapeutics. It offers an updated overview on the main results in the field and authors made efforts to highlight the main obstacles in the usage of certain approaches in research and clinics.

The manuscript is very well written and well structured, while the overall presentation is detailed, clear and informative. Future Directions are concise, while major conclusions correspond to the state of the art. Still, there are some corrections that need to be made:

- References 12, 13 and 14 are not adequate for the statements of page 2 (last paragraph). Instead of linking prostate microbiota to female bladder bacteria and the pitfalls in the diagnostics of urinary tract infections in small children, authors should consider using some more appropriate references for their claims, such as 10.1016/j.eururo.2017.03.029. Ref 12 has to be replaced in this paragraph, since it has no relation to prostate cancer at all.

- In Tables 1 and 2 the authors should include one more column with the numeration of references, so that it would be easier to connect names of the Authors with their articles.  

Author Response

Reviewer #1 comment:- References 12, 13 and 14 are not adequate for the statements of page 2 (last paragraph). Instead of linking prostate microbiota to female bladder bacteria and the pitfalls in the diagnostics of urinary tract infections in small children, authors should consider using some more appropriate references for their claims, such as 10.1016/j.eururo.2017.03.029. Ref 12 has to be replaced in this paragraph, since it has no relation to prostate cancer at all.

Reply: Thank you for your insightful comments and for directing our attention to the potential discrepancies related to references 12, 13, and 14 in our manuscript.

Upon re-evaluation of the cited references in the context of the statements made in the last paragraph of page 2, we have discovered an inadvertent misalignment of citations. This misalignment appears to have arisen due to technical issues with the ENDNOTE program, leading to incorrect linking and subsequent citation errors. We deeply regret this oversight and any confusion it may have caused. I have corrected the original citation to ensure the link is properly maintained. And to maintain an appropriate level of evidence, we have added references and modified the paragraph on page 2 as follows (now page3):

“Although the relationship between prostate cancer and urinary microbiome is not well-documented, the microbiome may be a key factor that induces chronic prostate inflammation. Infection and inflammation in the prostate, which are related to the prostatic microbiome, lead to microenvironmental changes in the prostate and promote prostatic carcinogenesis [1-3]. Previous studies have shown that the prostate microbiome is similar to that of the urethra, supporting bacterial translocation to the prostate through the urethra [4, 5]. In other words, asymptomatic chronic prostate inflammation can be induced by the urinary microbiome, which may be a notable cause of prostate cancer development. Intraprostatic urine reflux must be considered to further understand the relationship between urinary microbiome and prostatic inflammation [6-8]. Generally, men void 5–7 times daily, and every single voiding event spreads the urinary microbiome into the prostate [4, 5, 9]. Although the urine biomass is extremely low, these frequent daily events would cause asymptomatic prostatic microenvironmental changes and chronic inflammation. Furthermore, pre-existing urinary dysbiosis can cause prostatic dysbiosis [10, 11]. Although direct prostatic microbiome acquisition would be the best option for assessing the relationship between the microbiome and prostate cancer, such studies cannot be conducted easily because of their invasiveness and contamination risk. Considering these limitations, urine is one of the most easily accessible and reasonable specimens for microbiome research in prostate cancer.”

Reviewer #1: comments: - In Tables 1 and 2 the authors should include one more column with the numeration of references, so that it would be easier to connect names of the Authors with their articles.

Reply: We appreciate the suggestion and have added a column to Tables 1 and 2 to facilitate the connection between authors and their respective studies. The references are now conveniently placed after the second column.

Table 1. Recent studies of urinary microbiome in prostate cancer.

Authors  

Published Year,

Reference

Groups (n)

Urine sample

Study design

Main findings of pathogens

Haining Yu et al

2015, [30]

BPH (21)

PCa (13)

Expressed prostatic secretions

PCa vs. BPH

Bacteroidetes, Alphaproteobacteria, Firmicutes, Lachnospiraceae, Propionicimonas, Sphingomonas, Ochrobactrum

Eva Shrestha et al

2018, [31]

Non-PCa (65)

PCa (repeat Bx)(5)

Not mentioned

PCa vs. non-PC               

Anaerococcus lactolyticus, Varibaculum cambriense, Propionimicrobium lymphophyilum

Shaheen Alanee et al

2019, [32]

non-PCa (16)

PCa (14)  

First voided urine after prostate massage

PCa vs. non-PCa

VeillonellaStreptococcus, and Bacteroides (increased)

FaecalibacteriumLactobaccili, and Actinetobacter (decreased)

Kai-Yen Tsai et al

2022, [33]

BPH (77) 

nmPCa (59)

Control (36)

Mid-term voided urine

PCa vs Control

PCa vs BPH

Faecalibacterium, Staphylococcus,  Ruminococcaceaeu_UCG_002, Neisseria, Agathobacter, Alcaligenes, Pseudomonas, Lactobacillus, Akkermansia,

Cetobacterium

Rachel Hurst et al

2022, [34]

PCa w/u (300)

Hematuria (18)

First voided urine after DRE

PCa vs. non-PCa

(Risk/Prognosis)

Fenollaria, Fenollaria, Peptoniphilus, Prophyromonas, Anaerococcus, and Fusobacterium

PCa, Prostate cancer; DRE, Digital rectal examination; BPH, benign prostate hyperplasia; nmPCa, non-metastatic prostate cancer

Table 2. Recent studies of gut microbiome in prostate cancer.

Authors

Published year,

Reference

Study design

Groups (n)

Materials

Method

Main findings of pathogens

Liss et al.

2018, [62]

Prostate biopsy

PCa(64) vs. non-PCa(41)

Rectal swabs

16S rRNA amplicon sequencing

High in PCa: Bacteroides Streptococcus

Golombos et al.

2018, [63]

Prostate biopsy

PCa(20) vs. non-PCa(8)

Stool samples before

16S rDNA amplicon sequencing

High in PCa: Bacteroides massiliensis, Controls: Faecalibacterium prausnitzii, Eubacterium rectalie

Sfanos et al.

2018, [74]

PCa vs Control

NoMeds(16) vs. ADT(5) vs. AA

Rectal swabs

16S rDNA amplicon sequencing

High in ADT: Akkermansia muciniphila, Ruminococcaceae spp., Lachnospiraceae spp., AA: Brevibacteriaceae(low), Erysipelorichaceae(low), Streptococcaceae(low)

Alanee et al.

2019, [32]

Prostate biopsy

PCa(30) vs. non-PCa(16)

Rectal swabs

16S rRNA amplicon sequencing

High in PCa: Bacteroides

Daisley et al.

2020, [75]

PCa vs Control

ADT(21) vs. ADT+AA(14) vs. control(33)

Stool samples

16S rRNA amplicon sequencing

ADT: Corynebacterium (low)

High in ADT+AA: Akkermansia

Liu et al.

2020, [76]

PCa

HSPC(21) vs. CRPC(21)

Stool samples

16S rRNA amplicon sequencing

High in CRPC: Phascolarctobacterium and Ruminococcus

Matsushita et al.

2021, [64]

Prostate biopsy

PCa(30) vs. non-PCa(16)

Rectal swabs

16S rRNA amplicon sequencing

High in PCa: Rikenellaceae, Alistipes, and Lachnospira

Pernigoni et al.

2021, [77]

PCa

HSPC(19) vs. CRPC(55)

Rectal swabs

16S rDNA amplicon sequencing

High in HSPC: Prevotella stercorea

High in CRPC: Ruminococcus spp., Bacteroides spp.

PCa, prostate cancer; ADT, androgen depivation therapy; AA, Abiraterone acetate; HSPC, hormone sensitive prostate cancer; CRPC, castration-resistant prostate cancer; spp., species

References

  1. Sfanos, K. S., S. Yegnasubramanian, W. G. Nelson, and A. M. De Marzo. "The Inflammatory Microenvironment and Microbiome in Prostate Cancer Development." Nat Rev Urol 15, no. 1 (2018): 11-24.
  2. Porter, C. M., E. Shrestha, L. B. Peiffer, and K. S. Sfanos. "The Microbiome in Prostate Inflammation and Prostate Cancer." Prostate Cancer Prostatic Dis 21, no. 3 (2018): 345-54.
  3. Sfanos, K. S., and A. M. De Marzo. "Prostate Cancer and Inflammation: The Evidence." Histopathology 60, no. 1 (2012): 199-215.
  4. Dong, Q., D. E. Nelson, E. Toh, L. Diao, X. Gao, J. D. Fortenberry, and B. Van der Pol. "The Microbial Communities in Male First Catch Urine Are Highly Similar to Those in Paired Urethral Swab Specimens." PLoS One 6, no. 5 (2011): e19709.
  5. Nelson, D. E., Q. Dong, B. Van der Pol, E. Toh, B. Fan, B. P. Katz, D. Mi, R. Rong, G. M. Weinstock, E. Sodergren, and J. D. Fortenberry. "Bacterial Communities of the Coronal Sulcus and Distal Urethra of Adolescent Males." PLoS One 7, no. 5 (2012): e36298.
  6. Sfanos, K. S., J. Sauvageot, H. L. Fedor, J. D. Dick, A. M. De Marzo, and W. B. Isaacs. "A Molecular Analysis of Prokaryotic and Viral DNA Sequences in Prostate Tissue from Patients with Prostate Cancer Indicates the Presence of Multiple and Diverse Microorganisms." Prostate 68, no. 3 (2008): 306-20.
  7. Yow, M. A., S. N. Tabrizi, G. Severi, D. M. Bolton, J. Pedersen, G. G. Giles, and M. C. Southey. "Characterisation of Microbial Communities within Aggressive Prostate Cancer Tissues." Infect Agent Cancer 12 (2017): 4.
  8. Krieger, J. N., D. E. Riley, R. L. Vesella, D. C. Miner, S. O. Ross, and P. H. Lange. "Bacterial Dna Sequences in Prostate Tissue from Patients with Prostate Cancer and Chronic Prostatitis." J Urol 164, no. 4 (2000): 1221-8.
  9. Aragón, I. M., B. Herrera-Imbroda, M. I. Queipo-Ortuño, E. Castillo, J. S. Del Moral, J. Gómez-Millán, G. Yucel, and M. F. Lara. "The Urinary Tract Microbiome in Health and Disease." Eur Urol Focus 4, no. 1 (2018): 128-38.
  10. Jayalath, S., and D. Magana-Arachchi. "Dysbiosis of the Human Urinary Microbiome and Its Association to Diseases Affecting the Urinary System." Indian J Microbiol 62, no. 2 (2022): 153-66.
  11. Levy, M., A. A. Kolodziejczyk, C. A. Thaiss, and E. Elinav. "Dysbiosis and the Immune System." Nat Rev Immunol 17, no. 4 (2017): 219-32.

Reviewer 2 Report

Comments and Suggestions for Authors

Literature acquisition was conducted on PubMed and Google Scholar. In total how many articles were obtained and how were articles selected to be included for this review?

Please clarify the following sentence: “Traditionally, prostate cancer is affected by lifestyle factors, which alter the gut microbiome.” Does prostate cancer affect gut microbiome or lifestyle factors affect gut microbiome too?

Please check for consistencies in Table 2 – 16S rRNA or 16S DNA?

Author Response

Reviewer #2: Comments to author

Reviewer #2: comments: Literature acquisition was conducted on PubMed and Google Scholar. In total how many articles were obtained and how were articles selected to be included for this review?

Reply: Thank you for your valuable feedback and inquiry concerning our article. We appreciate the opportunity to clarify the methodology behind our literature acquisition and article selection process.

Our literature search was conducted systematically using the PubMed database and Google Scholar through October 2023. The keywords employed for this extensive search included combinations of the following terms: "microbiota", "microbiome", "cancer", "neoplasms", "prostate cancer", and "prostate neoplasm". This broad yet targeted approach was designed to encompass a wide range of studies pertinent to the role of microbiome in prostate cancer. This targeted approach yielded a substantial number of articles, specifically 322 for 'Microbiome Prostate cancer', 299 for 'Microbiota Prostate cancer', 112 for 'Microbiota Prostate neoplasm', and 137 for 'Microbiome Prostate neoplasm'.

Through this meticulous process, we compiled a core list of 150 articles and with their references that form the backbone of our review article. These selected articles were critically analyzed and synthesized to provide a comprehensive overview of the current understanding of the role of microbiome in prostate cancer, its diagnostic potential, therapeutic implications, and future directions. The selection was based on several criteria: relevance to the microbiome's role in prostate cancer, recency and novelty of the research, robust methodology, clear data presentation, impact and citation rate, and a diversity of perspectives. These articles were critically analyzed and synthesized to provide a comprehensive overview of the current understanding of the role of microbiome in prostate cancer, including its diagnostic potential, therapeutic implications, and future directions.

Reviewer #2: comments: Please clarify the following sentence: “Traditionally, prostate cancer is affected by lifestyle factors, which alter the gut microbiome.” Does prostate cancer affect gut microbiome or lifestyle factors affect gut microbiome too?

Reply: Thank you for your insightful comment.

Traditional research suggests lifestyle factors such as physical activity and diet directly influence prostate cancer. Some of these factors, including metabolic syndrome and dietary factors, are known to interact with the gut microbiome. Thus, their impact is believed to be both direct and mediated through changes in the gut microbiome. We apologize for any confusion caused by our phrasing and have clarified this relationship in “Figure 1. Risk factors including urinary and gut microbiome related to the development of prostate cancer and therapeutics of gut microbiome.”

Figure 1. Risk factors including urinary and gut microbiome related to the development of prostate cancer and therapeutics of gut microbiome.

Reviewer #2: comments: Please check for consistencies in Table 2 – 16S rRNA or 16S DNA?

Reply: Thank you for your insightful comment.

Following the reviewer's recommendation, we have consistently revised the contents of the Table2.

Table 2. Recent studies of gut microbiome in prostate cancer.

Authors

Published year,

Reference

Study design

Groups (n)

Materials

Method

Main findings of pathogens

Liss et al.

2018, [1]

Prostate biopsy

PCa(64) vs. non-PCa(41)

Rectal swabs

16S rRNA amplicon sequencing

High in PCa: Bacteroides Streptococcus

Golombos et al.

2018, [2]

Prostate biopsy

PCa(20) vs. non-PCa(8)

Stool samples before

16S rDNA amplicon sequencing

High in PCa: Bacteroides massiliensis, Controls: Faecalibacterium prausnitzii, Eubacterium rectalie

Sfanos et al.

2018, [3]

PCa vs Control

NoMeds(16) vs. ADT(5) vs. AA

Rectal swabs

16S rDNA amplicon sequencing

High in ADT: Akkermansia muciniphila, Ruminococcaceae spp., Lachnospiraceae spp., AA: Brevibacteriaceae(low), Erysipelorichaceae(low), Streptococcaceae(low)

Alanee et al.

2019, [4]

Prostate biopsy

PCa(30) vs. non-PCa(16)

Rectal swabs

16S rRNA amplicon sequencing

High in PCa: Bacteroides

Daisley et al.

2020, [5]

PCa vs Control

ADT(21) vs. ADT+AA(14) vs. control(33)

Stool samples

16S rRNA amplicon sequencing

ADT: Corynebacterium (low)

High in ADT+AA: Akkermansia

Liu et al.

2020, [6]

PCa

HSPC(21) vs. CRPC(21)

Stool samples

16S rRNA amplicon sequencing

High in CRPC: Phascolarctobacterium and Ruminococcus

Matsushita et al.

2021, [7]

Prostate biopsy

PCa(30) vs. non-PCa(16)

Rectal swabs

16S rRNA amplicon sequencing

High in PCa: Rikenellaceae, Alistipes, and Lachnospira

Pernigoni et al.

2021, [8]

PCa

HSPC(19) vs. CRPC(55)

Rectal swabs

16S rDNA amplicon sequencing

High in HSPC: Prevotella stercorea

High in CRPC: Ruminococcus spp., Bacteroides spp.

PCa, prostate cancer; ADT, androgen depivation therapy; AA, Abiraterone acetate; HSPC, hormone sensitive prostate cancer; CRPC, castration-resistant prostate cancer; spp., species

References

  1. Liss, M. A., J. R. White, M. Goros, J. Gelfond, R. Leach, T. Johnson-Pais, Z. Lai, E. Rourke, J. Basler, D. Ankerst, and D. P. Shah. "Metabolic Biosynthesis Pathways Identified from Fecal Microbiome Associated with Prostate Cancer." Eur Urol 74, no. 5 (2018): 575-82.
  2. Golombos, D. M., A. Ayangbesan, P. O'Malley, P. Lewicki, L. Barlow, C. E. Barbieri, C. Chan, C. DuLong, G. Abu-Ali, C. Huttenhower, and D. S. Scherr. "The Role of Gut Microbiome in the Pathogenesis of Prostate Cancer: A Prospective, Pilot Study." Urology 111 (2018): 122-28.
  3. Sfanos, K. S., M. C. Markowski, L. B. Peiffer, S. E. Ernst, J. R. White, K. J. Pienta, E. S. Antonarakis, and A. E. Ross. "Compositional Differences in Gastrointestinal Microbiota in Prostate Cancer Patients Treated with Androgen Axis-Targeted Therapies." Prostate Cancer Prostatic Dis 21, no. 4 (2018): 539-48.
  4. Alanee, S., A. El-Zawahry, D. Dynda, A. Dabaja, K. McVary, M. Karr, and A. Braundmeier-Fleming. "A Prospective Study to Examine the Association of the Urinary and Fecal Microbiota with Prostate Cancer Diagnosis after Transrectal Biopsy of the Prostate Using 16srna Gene Analysis." Prostate 79, no. 1 (2019): 81-87.
  5. Daisley, B. A., R. M. Chanyi, K. Abdur-Rashid, K. F. Al, S. Gibbons, J. A. Chmiel, H. Wilcox, G. Reid, A. Anderson, M. Dewar, S. M. Nair, J. Chin, and J. P. Burton. "Abiraterone Acetate Preferentially Enriches for the Gut Commensal Akkermansia Muciniphila in Castrate-Resistant Prostate Cancer Patients." Nat Commun 11, no. 1 (2020): 4822.
  6. Liu, Y., and H. Jiang. "Compositional Differences of Gut Microbiome in Matched Hormone-Sensitive and Castration-Resistant Prostate Cancer." Transl Androl Urol 9, no. 5 (2020): 1937-44.
  7. Matsushita, M., K. Fujita, D. Motooka, K. Hatano, S. Fukae, N. Kawamura, E. Tomiyama, Y. Hayashi, E. Banno, T. Takao, S. Takada, S. Yachida, H. Uemura, S. Nakamura, and N. Nonomura. "The Gut Microbiota Associated with High-Gleason Prostate Cancer." Cancer Sci 112, no. 8 (2021): 3125-35.
  8. Pernigoni, N., E. Zagato, A. Calcinotto, M. Troiani, R. P. Mestre, B. Calì, G. Attanasio, J. Troisi, M. Minini, S. Mosole, A. Revandkar, E. Pasquini, A. R. Elia, D. Bossi, A. Rinaldi, P. Rescigno, P. Flohr, J. Hunt, A. Neeb, L. Buroni, C. Guo, J. Welti, M. Ferrari, M. Grioni, J. Gauthier, R. Z. Gharaibeh, A. Palmisano, G. M. Lucchini, E. D'Antonio, S. Merler, M. Bolis, F. Grassi, A. Esposito, M. Bellone, A. Briganti, M. Rescigno, J. P. Theurillat, C. Jobin, S. Gillessen, J. de Bono, and A. Alimonti. "Commensal Bacteria Promote Endocrine Resistance in Prostate Cancer through Androgen Biosynthesis." Science 374, no. 6564 (2021): 216-24.

Reviewer 3 Report

Comments and Suggestions for Authors

The authors have attempted to discuss the potential of gut microbiome modification through dietary interventions, prebiotics, probiotics, and fecal microbiota transplantation for improving treatment response and mitigating adverse effects in patients with prostate cancer. Moreover, this review discusses the potential of microbiome profiling for patient stratification and personalized treatment strategies.

The study is methodologically well performed.

There are a few of points which may be considered for further improvement.

Please provide reference justifying asymptomatic chronic prostate inflammation can be induced by the urinary microbiome, which may be a notable cause of prostate cancer development.” Section 3.1

Please provide reference justifying men void 5–7 times daily, and every single voiding event spreads the urinary microbiome into the prostate. Although the urine biomass is extremely low, these frequent daily events would cause asymptomatic prostatic microenvironmental changes and chronic inflammation. Section 3.1

Please list the limitations of this study at the end of the discussion section, if there are any.

Author Response

Reviewer #3: Comments to author

The authors have attempted to discuss the potential of gut microbiome modification through dietary interventions, prebiotics, probiotics, and fecal microbiota transplantation for improving treatment response and mitigating adverse effects in patients with prostate cancer. Moreover, this review discusses the potential of microbiome profiling for patient stratification and personalized treatment strategies.

The study is methodologically well performed.

There are a few of points which may be considered for further improvement.

Reviewer #3: Please provide reference justifying „asymptomatic chronic prostate inflammation can be induced by the urinary microbiome, which may be a notable cause of prostate cancer development.” Section 3.1

Reply: Thank you for your insightful comment, reviewer.

We regret providing insufficient references and have addressed this by adding additional citations to the paragraph on page 3 to enhance the substantiation level. The revised content, including the newly added references, is as follows:

“Although the relationship between prostate cancer and urinary microbiome is not well-documented, the microbiome may be a key factor that induces chronic prostate inflammation. Infection and inflammation in the prostate, which are related to the prostatic microbiome, lead to microenvironmental changes in the prostate and promote prostatic carcinogenesis [1-3]. Previous studies have shown that the prostate microbiome is similar to that of the urethra, supporting bacterial translocation to the prostate through the urethra [4, 5]. In other words, asymptomatic chronic prostate inflammation can be induced by the urinary microbiome, which may be a notable cause of prostate cancer development. Intraprostatic urine reflux must be considered to further understand the relationship between urinary microbiome and prostatic inflammation [6-8]. Generally, men void 5–7 times daily, and every single voiding event spreads the urinary microbiome into the prostate [4, 5, 9]. Although the urine biomass is extremely low, these frequent daily events would cause asymptomatic prostatic microenvironmental changes and chronic inflammation. Furthermore, pre-existing urinary dysbiosis can cause prostatic dysbiosis [10, 11]. Although direct prostatic microbiome acquisition would be the best option for assessing the relationship between the microbiome and prostate cancer, such studies cannot be conducted easily because of their invasiveness and contamination risk. Considering these limitations, urine is one of the most easily accessible and reasonable specimens for microbiome research in prostate cancer.”

Reviewer #3: Please provide reference justifying „men void 5–7 times daily, and every single voiding event spreads the urinary microbiome into the prostate. Although the urine biomass is extremely low, these frequent daily events would cause asymptomatic prostatic microenvironmental changes and chronic inflammation.” Section 3.1

Reply: Thank you for your insightful comment, reviewer.

We regret providing insufficient references and have addressed this by adding additional citations to the paragraph on page 3 to enhance the substantiation level. The revised content, including the newly added references, is as follows:

“Although the relationship between prostate cancer and urinary microbiome is not well-documented, the microbiome may be a key factor that induces chronic prostate inflammation. Infection and inflammation in the prostate, which are related to the prostatic microbiome, lead to microenvironmental changes in the prostate and promote prostatic carcinogenesis [1-3]. Previous studies have shown that the prostate microbiome is similar to that of the urethra, supporting bacterial translocation to the prostate through the urethra [4, 5]. In other words, asymptomatic chronic prostate inflammation can be induced by the urinary microbiome, which may be a notable cause of prostate cancer development. Intraprostatic urine reflux must be considered to further understand the relationship between urinary microbiome and prostatic inflammation [6-8]. Generally, men void 5–7 times daily, and every single voiding event spreads the urinary microbiome into the prostate [4, 5, 9]. Although the urine biomass is extremely low, these frequent daily events would cause asymptomatic prostatic microenvironmental changes and chronic inflammation. Furthermore, pre-existing urinary dysbiosis can cause prostatic dysbiosis [10, 11]. Although direct prostatic microbiome acquisition would be the best option for assessing the relationship between the microbiome and prostate cancer, such studies cannot be conducted easily because of their invasiveness and contamination risk. Considering these limitations, urine is one of the most easily accessible and reasonable specimens for microbiome research in prostate cancer.”

Reviewer #3: Please list the limitations of this study at the end of the discussion section, if there are any.

Reply: Thank you for your insightful comment, reviewer.

We have recognized the inadequacies in our discussion regarding the limitations of microbiome research and have accordingly composed an additional discussion to address this matter in page 10 :

“There are several limitations to urinary and gut microbiome studies. In urinary microbiome research, studies have primarily focused on the differences between prostate cancer patients and non-prostate cancer patients. However, the results concerning the types of identified microbiomes are often inconsistent. This inconsistency may stem from varied study methodologies, including urine collection and processing techniques, which can lead to contamination and affect the outcomes. Moreover, the causal relationship between urinary microbiome alterations and prostate cancer development remains unclear. The temporal gap between the typical age of occurrence of urinary tract infections and the age of prostate cancer onset further complicates our understanding of any potential causal links, suggesting a need for prospective longitudinal studies to elucidate this relationship. Regarding the gut microbiome, research has primarily focused on the differences between prostate cancer patients and those without, as well as between high-risk and non-prostate cancer patients. This approach is a principal method for understanding the complex interplay between the gut microbiome and prostate cancer, but many aspects of the causal relationships remain to be elucidated. Recent studies have indicated that the gut microbiome might regulate testosterone levels, which are crucial in prostate cancer treatment. These findings suggest that the gut microbiome interferes with testosterone deprivation and could be linked to treatment failures. Although the evidence from these studies provides a basis for understanding the causal links and FMT has shown potential as a treatment method in these experimental settings, its effectiveness typically requires continuous application. Therefore, the real-world efficacy of other possible treatments needs further exploration. Furthermore, the use of prebiotics and probiotics necessitates additional research involving human subjects to validate their therapeutic potential and establish a more concrete basis for their use in managing prostate cancer.”

References

  1. Sfanos, K. S., S. Yegnasubramanian, W. G. Nelson, and A. M. De Marzo. "The Inflammatory Microenvironment and Microbiome in Prostate Cancer Development." Nat Rev Urol 15, no. 1 (2018): 11-24.
  2. Porter, C. M., E. Shrestha, L. B. Peiffer, and K. S. Sfanos. "The Microbiome in Prostate Inflammation and Prostate Cancer." Prostate Cancer Prostatic Dis 21, no. 3 (2018): 345-54.
  3. Sfanos, K. S., and A. M. De Marzo. "Prostate Cancer and Inflammation: The Evidence." Histopathology 60, no. 1 (2012): 199-215.
  4. Dong, Q., D. E. Nelson, E. Toh, L. Diao, X. Gao, J. D. Fortenberry, and B. Van der Pol. "The Microbial Communities in Male First Catch Urine Are Highly Similar to Those in Paired Urethral Swab Specimens." PLoS One 6, no. 5 (2011): e19709.
  5. Nelson, D. E., Q. Dong, B. Van der Pol, E. Toh, B. Fan, B. P. Katz, D. Mi, R. Rong, G. M. Weinstock, E. Sodergren, and J. D. Fortenberry. "Bacterial Communities of the Coronal Sulcus and Distal Urethra of Adolescent Males." PLoS One 7, no. 5 (2012): e36298.
  6. Sfanos, K. S., J. Sauvageot, H. L. Fedor, J. D. Dick, A. M. De Marzo, and W. B. Isaacs. "A Molecular Analysis of Prokaryotic and Viral DNA Sequences in Prostate Tissue from Patients with Prostate Cancer Indicates the Presence of Multiple and Diverse Microorganisms." Prostate 68, no. 3 (2008): 306-20.
  7. Yow, M. A., S. N. Tabrizi, G. Severi, D. M. Bolton, J. Pedersen, G. G. Giles, and M. C. Southey. "Characterisation of Microbial Communities within Aggressive Prostate Cancer Tissues." Infect Agent Cancer 12 (2017): 4.
  8. Krieger, J. N., D. E. Riley, R. L. Vesella, D. C. Miner, S. O. Ross, and P. H. Lange. "Bacterial Dna Sequences in Prostate Tissue from Patients with Prostate Cancer and Chronic Prostatitis." J Urol 164, no. 4 (2000): 1221-8.
  9. Aragón, I. M., B. Herrera-Imbroda, M. I. Queipo-Ortuño, E. Castillo, J. S. Del Moral, J. Gómez-Millán, G. Yucel, and M. F. Lara. "The Urinary Tract Microbiome in Health and Disease." Eur Urol Focus 4, no. 1 (2018): 128-38.
  10. Jayalath, S., and D. Magana-Arachchi. "Dysbiosis of the Human Urinary Microbiome and Its Association to Diseases Affecting the Urinary System." Indian J Microbiol 62, no. 2 (2022): 153-66.
  11. Levy, M., A. A. Kolodziejczyk, C. A. Thaiss, and E. Elinav. "Dysbiosis and the Immune System." Nat Rev Immunol 17, no. 4 (2017): 219-32.

Reviewer 4 Report

Comments and Suggestions for Authors

This study attempts to addresses the correlation between microbiome and prostate cancer, and further explores the diagnostic and therapeutic implications. The potential of gut microbiome modification through dietary interventions, prebiotics, probiotics, and fecal microbiota transplantation are reviewed for improving treatment response and mitigating adverse effects. Moreover, the study concludes that the role of urinary and gut microbiomes in prostate cancer improves our understanding of cancer pathogenesis and management. And the potential of urinary microbiome as a non-invasive biomarker allows early detection and risk assessment in prostate cancer, and so on. If true, these results are very helpful to the diagnosis and treatment of prostate diseases and related diseases. Unfortunately, these results may seem uncertain at the moment.

   The main shortcomings of this paper include as follows

 1) it lacks quantitative comparative analysis, and the conclusions obtained solely through data statistics are difficult to convince;

2) In terms of the development pattern of pathology, the study does not make much contribution because the full text does not see the correlation between the results and time development;

3) The classification of pathology is also incomplete, and the information currently available is too limited.

Comments on the Quality of English Language

OK

Author Response

Reviewer #4: comment: it lacks quantitative comparative analysis, and the conclusions obtained solely through data statistics are difficult to convince;

Reply: Thank you for your insightful comment.

The critique highlights a crucial aspect of empirical research—the robustness and persuasiveness of conclusions drawn from quantitative analyses. While our review primarily focuses on synthesizing existing literature and identifying potential diagnostic and therapeutic implications of the microbiome in prostate cancer, we acknowledge the limitation stemming from the lack of primary quantitative comparative analysis. It is indeed paramount for any review to not only summarize findings but also to critically assess and compare them quantitatively where possible.

To address this, we propose future directions where rigorous meta-analyses and systematic reviews can be conducted to quantitatively compare the results of different studies. This approach would provide a more robust statistical backing to the observations and hypotheses presented. Furthermore, encouraging primary research with standardized methodologies would also contribute to building a more comparative and quantitative understanding of the role of microbiome in prostate cancer.

Reviewer #4: comment: In terms of the development pattern of pathology, the study does not make much contribution because the full text does not see the correlation between the results and time development;

Reply: We appreciate your observation on the importance of temporal development in understanding the pathology of prostate cancer. The dynamic nature of the human microbiome and its interaction with prostate cancer over time is indeed a critical aspect that warrants deeper exploration. Our review primarily collates cross-sectional insights, which limits the ability to infer the temporal development of pathology.

To mitigate this, we suggest that future research should focus more on longitudinal study designs. These studies can track changes in the urinary and gut microbiomes over time and correlate these changes with the progression or remission of prostate cancer. Understanding the temporal dynamics will significantly contribute to elucidating the causal relationships and potentially pave the way for predictive diagnostics and targeted interventions. We added it limitations

“There are several limitations to urinary and gut microbiome studies. In urinary microbiome research, studies have primarily focused on the differences between prostate cancer patients and non-prostate cancer patients. However, the results concerning the types of identified microbiomes are often inconsistent. This inconsistency may stem from varied study methodologies, including urine collection and processing techniques, which can lead to contamination and affect the outcomes. Moreover, the causal relationship between urinary microbiome alterations and prostate cancer development remains unclear. The temporal gap between the typical age of occurrence of urinary tract infections and the age of prostate cancer onset further complicates our understanding of any potential causal links, suggesting a need for prospective longitudinal studies to elucidate this relationship. Regarding the gut microbiome, research has primarily focused on the differences between prostate cancer patients and those without, as well as between high-risk and non-prostate cancer patients. This approach is a principal method for understanding the complex interplay between the gut microbiome and prostate cancer, but many aspects of the causal relationships remain to be elucidated. Recent studies have indicated that the gut microbiome might regulate testosterone levels, which are crucial in prostate cancer treatment. These findings suggest that the gut microbiome interferes with testosterone deprivation and could be linked to treatment failures. Although the evidence from these studies provides a basis for understanding the causal links and FMT has shown potential as a treatment method in these experimental settings, its effectiveness typically requires continuous application. Therefore, the real-world efficacy of other possible treatments needs further exploration. Furthermore, the use of prebiotics and probiotics necessitates additional research involving human subjects to validate their therapeutic potential and establish a more concrete basis for their use in managing prostate cancer.”

Reviewer #4: comment: The classification of pathology is also incomplete, and the information currently available is too limited.

Reply: Your critique rightly points out the challenges in the comprehensive classification of pathology due to the limited scope of current research. Prostate cancer is a complex disease with multiple influencing factors, including the microbiome. In response to this, we emphasize the need for an integrative classification framework that incorporates microbial profiles. As research in this domain progresses, a more nuanced understanding of how microbial variations influence different prostate cancer pathologies will emerge. Collaborative efforts between urologists, oncologists, and microbiologists will be crucial in developing a more holistic and detailed classification system that includes microbiome data. This will not only enhance our understanding of prostate cancer pathology but also aid in personalized treatment strategies.

Round 2

Reviewer 4 Report

Comments and Suggestions for Authors

I have reviewed the author's cover letter and the opinions of other reviewers, as well as reviewed all the revisions. I have changed my original decision to agree to accept this article, mainly based on the following two points: 1) Indeed, there is very little research on the prediction and analysis of diseases such as the prostate, and the samples are not easily obtained; 2) The results presented in this article are somewhat innovative. Perhaps it can reach the average level of this journal.

Comments on the Quality of English Language

OK

Author Response

Reviewer 4 2nd revision

Reviewer #4 comment:- I have reviewed the author's cover letter and the opinions of other reviewers, as well as reviewed all the revisions. I have changed my original decision to agree to accept this article, mainly based on the following two points: 1) Indeed, there is very little research on the prediction and analysis of diseases such as the prostate, and the samples are not easily obtained; 2) The results presented in this article are somewhat innovative. Perhaps it can reach the average level of this journal.

Reply: Thank you for insightful comment.

Building upon the reviewer's constructive feedback, our study delves deeper into the innovative and intricate relationship between the microbiome and cancer. We explore how recent research not only introduces new perspectives but also provides comprehensive insights into the significant impact of host-microbiome interactions.

Reply 1: In response to the reviewer's insightful comments and the recognized scarcity of research in the prediction and analysis of prostate-related diseases, we have augmented our introduction to further emphasize the innovative nature and significance of our study in filling this critical research gap. The added introduction (Section 1) is as follows:

Prostate cancer remains a significant global health challenge. Its etiology and progression are influenced by various factors, including genetics, lifestyle, and, increasingly, the microbiome. Despite the known impact of the microbiome on health and disease, its role in prostate cancer remains under-explored. While recent research has begun to focus on the microbiome's role in prostate cancer, the practical applications of these findings in diagnosis and treatment have not been sufficiently elucidated. This oversight highlights the need for an increased focus on how the microbiome can be integrated into clinical practice. In this review, we explore the potential uses of pharmaceutics and examine the limitations of urinary and gut microbiomes in various aspects of prostate cancer management, including prevention, diagnosis, risk stratification, and treatment. Our investigation aligns with recent calls for more in-depth research in this domain.

Reply 2: In response to the constructive feedback and suggestions from the reviewers, we have carefully expanded our manuscript to further elaborate on the potential role of the microbiome as a biomarker for the analysis and prediction of prostate cancer. The added (Section 4.4) is as follows:

Several epidemiological studies have indicated a correlation between specific urinary and gut microbiota compositions and an increased risk of prostate cancer. Additionally, various studies have presented mechanistic evidence linking the microbiome to prostate cancer, suggesting its potential as a biomarker. The role of gut microbiota in endocrine resistance in prostate cancer has been substantiated, at least partially, through the generation of androgenic steroids that facilitate carcinogenic signaling via androgen receptors [1]. The production of bacterial genotoxins is associated with the occurrence of the TMPRSS2-ERG gene fusion, a common early carcinogenic event during the development of prostate cancer. Therefore, the host microbiota provides mechanistic evidence relating to the onset and progression of prostate cancer, suggesting that profiling and modulation of the host microbiome could offer potential avenues for detection and treatment [2]. Intriguingly, the gut microbiota profile itself has emerged as a novel and useful marker for the detection of high-risk prostate cancer, further implicating the microbiome in the carcinogenesis of prostate cancer [3]. Alterations in microbiota composition have been identified as essential factors in the occurrence, development, and prognosis of prostate cancer, thus reinforcing the microbiome's potential as an innovative biomarker [4]. The gut microbiota is involved in the progression and chemoresistance of prostate cancer and provides insights into the mechanisms of the microbiota's influence and potentials as a biomarker for prostate cancer [5]. This association elucidates the link between the gut microbiome and prostate cancer, emphasizing how a high-fat diet leads to gut dysbiosis and promotes prostate cancer growth through bacterial metabolites [6]. Comprehensive studies profiling various components of the microbiome—including prostate tissue, gut, urinary, seminal fluid, and prostatic fluid—in relation to prostate cancer provide an extensive understanding of the diverse microbiome compositions and their associations with the disease. Such studies are invaluable in reinforcing the potential of the microbiome as a biomarker and in delineating the multifaceted roles of microorganisms in prostate cancer pathogenesis [7]. Additionally, the presence of pro-inflammatory or pathogenic bacteria, particularly those involved in urinary tract pathologies, has been linked to the progression of prostate cancer. This highlights another facet of the role of the microbiome in prostate cancer, further endorsing its potential as a biomarker and a target for therapeutic strategies [8].

Reply 3: In alignment with the reviewer's insightful commentary on enhancing the discussion on innovative therapeutic opportunities, we have incorporated recent advancements and elaborated on how our research contributes to this field. The added (Section 4.4) is as follows:

Innovative therapeutic strategies are crucial for advancing prostate cancer treatment. Recent studies have explored the potential of microbiome modification through various means as a therapeutic strategy. This emerging field proposes several treatment strategies, including specific dietary interventions, probiotics, prebiotics, and fecal microbiota transplantation, aimed at improving treatment responses and personalizing therapy for prostate cancer patients. One study explores the use of probiotics as a potential adjuvant therapy for prostate cancer, suggesting that probiotics might enhance the response to treatment and reduce the risk of post-operative infections. This indicates the potential of microbiome-related strategies in prostate cancer therapy [9]. Additionally, research has examined the impact of the gut microbiome on the therapeutic efficacy of prostate cancer immunotherapies, including cyclophosphamide, anti-PD-L1, and CTLA-4 blockade. This suggests that the microbiome may serve as a therapy target to be modulated, offering a potential therapeutic strategy for prostate cancer [10]. Furthermore, the emerging evidence of the microbiome's role in health and disease suggests that alterations in microbiota composition (dysbiosis) may play a crucial role in the occurrence, development, and prognosis of prostate cancer. This highlights the potential of microbiome modification as a novel therapeutic strategy [4].

Reply 4: In response to the reviewer's comments emphasizing the importance of discussing the potential of the microbiome as a non-invasive biomarker for the early detection and risk assessment of prostate cancer, we have crafted the following text (Section 4.1):

The identification of reliable, non-invasive biomarkers for early detection and risk assessment in prostate cancer is valuable. The urinary and gut microbiomes hold significant potential in this regard, as recent research supports the microbiome's diagnostic capabilities. Growing knowledge about the relationship between the microbiome and prostate cancer reveals that cancerous prostate tissue contains bacterial DNA, unlike healthy prostate tissue. The urinary microbiome, when linked to chronic inflammation, has been associated with an increased risk of prostate cancer and its pathogenesis, suggesting its potential as a diagnostic tool [11]. Several studies have focused on the differences in the microbiota of patients with prostate cancer. Dysbiosis may affect the inflammatory status, hormone levels, and microbial metabolites, leading to prostate cancer progression. This provides a basis for the role of the microbiome as a biomarker for early detection and assessing prostate cancer risk [12]. Additionally, the role of the gut microbiome in prostate cancer endocrine resistance, which occurs through the generation of androgenic steroids, indicates a possible diagnostic capability of the microbiome. Its composition could potentially reveal the endocrine resistance status of a prostate cancer patient [13].

Reply 5: In response to the reviewer's comments, the innovative insights offered, especially regarding diagnostic and therapeutic potential, are set to significantly impact the field. We have crafted the following section (section 4.5):

In recent research, perspectives on the microbiome are not only innovative but also providing comprehensive insights from a viewpoint. These perspectives emphasize the significant impact of host-microbiome interactions on multiple physiological processes and a range of multifactorial disease conditions [14-18]. The role of commensal bacteria and other microorganisms that colonize the epithelial surfaces of the body is a subject of intense study. The production of small molecules and metabolites by these microorganisms underscores the intricate relationship between the microbiome and cancer, necessitating further exploration [18]. The literature frequently discusses host mechanisms by which the microbiome may modify carcinogenesis, thus offering a comprehensive understanding of the microbiome's role in cancer and its potential implications for diagnosis and therapy [14]. Additionally, the use of the salivary microbiome, fecal microbiome, and circulating microbial DNA in blood plasma as experimental diagnostic biomarkers for numerous types of cancer supports the potential diagnostic capabilities of the microbiome [15]. Furthermore, the human microbiome, recognized as a complex community, engages in symbiotic interactions with the host across multiple body sites [16]. The impact of microorganisms on cancer immunotherapy proposes a new direction for improving the effects of tumor immunotherapy through the application of microorganisms. It provides insights into carcinogenic microorganism infections, microbial disorders, and carcinogenesis, offering a comprehensive view of the role of the microbiome [17].

  1. Pernigoni, N., E. Zagato, A. Calcinotto, M. Troiani, R. P. Mestre, B. Calì, G. Attanasio, J. Troisi, M. Minini, S. Mosole, A. Revandkar, E. Pasquini, A. R. Elia, D. Bossi, A. Rinaldi, P. Rescigno, P. Flohr, J. Hunt, A. Neeb, L. Buroni, C. Guo, J. Welti, M. Ferrari, M. Grioni, J. Gauthier, R. Z. Gharaibeh, A. Palmisano, G. M. Lucchini, E. D'Antonio, S. Merler, M. Bolis, F. Grassi, A. Esposito, M. Bellone, A. Briganti, M. Rescigno, J. P. Theurillat, C. Jobin, S. Gillessen, J. de Bono, and A. Alimonti. "Commensal Bacteria Promote Endocrine Resistance in Prostate Cancer through Androgen Biosynthesis." Science 374, no. 6564 (2021): 216-24.
  2. Pernigoni, Nicolò, Christina Guo, Lewis Gallagher, Wei Yuan, Manuel Colucci, Martina Troiani, Lei Liu, Luisa Maraccani, Ilaria Guccini, Denis Migliorini, Johann de Bono, and Andrea Alimonti. "The Potential Role of the Microbiota in Prostate Cancer Pathogenesis and Treatment." Nature Reviews Urology 20, no. 12 (2023): 706-18.
  3. Matsushita, M., K. Fujita, D. Motooka, K. Hatano, S. Fukae, N. Kawamura, E. Tomiyama, Y. Hayashi, E. Banno, T. Takao, S. Takada, S. Yachida, H. Uemura, S. Nakamura, and N. Nonomura. "The Gut Microbiota Associated with High-Gleason Prostate Cancer." Cancer Sci 112, no. 8 (2021): 3125-35.
  4. Kustrimovic, N., R. Bombelli, D. Baci, and L. Mortara. "Microbiome and Prostate Cancer: A Novel Target for Prevention and Treatment." Int J Mol Sci 24, no. 2 (2023).
  5. Zhong, W., K. Wu, Z. Long, X. Zhou, C. Zhong, S. Wang, H. Lai, Y. Guo, D. Lv, J. Lu, and X. Mao. "Gut Dysbiosis Promotes Prostate Cancer Progression and Docetaxel Resistance Via Activating Nf-Κb-Il6-Stat3 Axis." Microbiome 10, no. 1 (2022): 94.
  6. Fujita, K., M. Matsushita, E. Banno, M. A. De Velasco, K. Hatano, N. Nonomura, and H. Uemura. "Gut Microbiome and Prostate Cancer." Int J Urol 29, no. 8 (2022): 793-98.
  7. Salachan, P. V., and K. D. Sørensen. "Dysbiotic Microbes and How to Find Them: A Review of Microbiome Profiling in Prostate Cancer." J Exp Clin Cancer Res 41, no. 1 (2022): 31.
  8. Perez-Carrasco, V., A. Soriano-Lerma, M. Soriano, J. Gutiérrez-Fernández, and J. A. Garcia-Salcedo. "Urinary Microbiome: Yin and Yang of the Urinary Tract." Front Cell Infect Microbiol 11 (2021): 617002.
  9. Garbas, K., P. Zapała, Ł Zapała, and P. Radziszewski. "The Role of Microbial Factors in Prostate Cancer Development-an up-to-Date Review." J Clin Med 10, no. 20 (2021).
  10. Fang, C., L. Wu, C. Zhu, W. Z. Xie, H. Hu, and X. T. Zeng. "A Potential Therapeutic Strategy for Prostatic Disease by Targeting the Oral Microbiome." Med Res Rev 41, no. 3 (2021): 1812-34.
  11. Katongole, P., O. J. Sande, M. Joloba, S. J. Reynolds, and N. Niyonzima. "The Human Microbiome and Its Link in Prostate Cancer Risk and Pathogenesis." Infect Agent Cancer 15 (2020): 53.
  12. Xia, B., J. Wang, D. Zhang, and X. Hu. "The Human Microbiome Links to Prostate Cancer Risk and Treatment (Review)." Oncol Rep 49, no. 6 (2023).
  13. Pernigoni, N., C. Guo, L. Gallagher, W. Yuan, M. Colucci, M. Troiani, L. Liu, L. Maraccani, I. Guccini, D. Migliorini, J. de Bono, and A. Alimonti. "The Potential Role of the Microbiota in Prostate Cancer Pathogenesis and Treatment." Nat Rev Urol 20, no. 12 (2023): 706-18.
  14. Queen, J., F. Shaikh, and C. L. Sears. "Understanding the Mechanisms and Translational Implications of the Microbiome for Cancer Therapy Innovation." Nat Cancer 4, no. 8 (2023): 1083-94.
  15. Kandalai, S., H. Li, N. Zhang, H. Peng, and Q. Zheng. "The Human Microbiome and Cancer: A Diagnostic and Therapeutic Perspective." Cancer Biol Ther 24, no. 1 (2023): 2240084.
  16. Cullin, N., C. Azevedo Antunes, R. Straussman, C. K. Stein-Thoeringer, and E. Elinav. "Microbiome and Cancer." Cancer Cell 39, no. 10 (2021): 1317-41.
  17. Zhou, P., Y. Hu, X. Wang, L. Shen, X. Liao, Y. Zhu, J. Yu, F. Zhao, Y. Zhou, H. Shen, and J. Li. "Microbiome in Cancer: An Exploration of Carcinogenesis, Immune Responses and Immunotherapy." Front Immunol 13 (2022): 877939.
  18. Sepich-Poore, G. D., L. Zitvogel, R. Straussman, J. Hasty, J. A. Wargo, and R. Knight. "The Microbiome and Human Cancer." Science 371, no. 6536 (2021).
